# UCB Exploration for Fixed-Budget Bayesian Best Arm Identification

**Rong J.B. Zhu**                                                   *rongzhu@fudan.edu.cn*
*Institute of Science and Technology for Brain-inspired Intelligence, Fudan University*

**Yanqi Qiu**                                                       *yanqi.qiu@hotmail.com*
*School of Mathematics and Statistics, Wuhan University*

**Reviewed on OpenReview:** *https://openreview.net/forum?id=BqSi73krYd*

## Abstract

We study best-arm identification (BAI) in the fixed-budget setting. Adaptive allocations based on upper confidence bounds (UCBs), such as `UCBE`, are known to work well in BAI. However, it is well-known that its optimal regret is theoretically dependent on instances, which we show to be an artifact in many fixed-budget BAI problems. In this paper we propose an UCB exploration algorithm that is both theoretically and empirically efficient for the fixed budget BAI problem under a Bayesian setting. The key idea is to learn prior information, which can enhance the performance of UCB-based BAI algorithm as it has done in the cumulative regret minimization problem. We establish bounds on the failure probability and the simple regret for the Bayesian BAI problem, providing upper bounds of order $\tilde{O}(\sqrt{K/n})$, up to logarithmic factors, where $n$ represents the budget and $K$ denotes the number of arms. Furthermore, we demonstrate through empirical results that our approach consistently outperforms state-of-the-art baselines.

## 1 Introduction

We study best-arm identification (BAI) in stochastic multi-armed bandits (Audibert et al., 2010; Karnin et al., 2013; Even-Dar et al., 2006; Bubeck et al., 2009; Jamieson et al., 2014; Kaufmann et al., 2015). In this problem, the learning agent sequentially interacts with the environment by pulling arms and receiving their rewards, which are sampled i.i.d. from their distributions. At the end, the agent must commit to a single arm. In the standard bandit setting, the agent maximizes its cumulative reward (Lai & Robbins, 1985; Auer et al., 2002; Lattimore & Szepesvári, 2019; Zhu & Rigotti, 2021). In fixed-budget BAI (Audibert et al., 2010; Karnin et al., 2013; Jamieson & Talwalkar, 2015; Li et al., 2018), the agent maximizes the probability of choosing the best arm within a fixed budget. In fixed-confidence BAI, the agent minimizes the budget to attain a target confidence level for identifying the best arm (Even-Dar et al., 2006; Audibert et al., 2010; Karnin et al., 2013). Here we focus on fixed-budget BAI.

Adaptive allocations based on upper confidence bounds (UCBs) are known to work well in fixed-budget BAI. For example, `UCBE` (Audibert et al., 2010) is optimal, with failure probability decreasing exponentially up to logarithmic factors. However, it relies on a plug-in approach of an unknown problem complexity term, learning to the adaptive variant performing significantly worse (Karnin et al., 2013). As a result, phase-based algorithms with uniform exploration in each phase, such as *successive rejects (SR)* (Audibert et al., 2010) and *sequential halving (SH)* (Karnin et al., 2013), have been shown to work better in practice. Furthermore, it should be noted that irrespective of the choice of the algorithms, i.e., UCB-based algorithms or phase-based algorithms, the optimal regret that decays exponentially are conditioned on the gaps between the maximal arm and the other arms not being small. If the gaps are small, the regret may decay polynomially instead of exponentially, as we will demonstrate in the next section.

It is well known that side information, such as the prior distribution of arm means, can improve the statistical efficiency of the cumulative regret minimization problem (Thompson, 1933; Chapelle & Li, 2011; Zhu & Kveton, 2022a). Motivated by this, we propose a novel, theoretically and empirically efficient, and instance-independent UCB exploration algorithm for identifying the best arm by learning the prior information of arm means. We consider a Bayesian prior setting on arm means, where arm means are sampled i.i.d. from a Gaussian distribution, with mean $\mu_0$ and variance $\sigma_0^2$. The mean $\mu_0$ is shared among the arms. The variance $\sigma_0^2$ characterizes the spread of the arms. A lower $\sigma_0^2$ means that the optimal arm is harder to identify, since the gaps between the optimal and suboptimal arms are smaller on average. Our study shows that learning the prior of arm means also improves the performance of the UCB-based BAI algorithm and makes it more practical, as it has done in the cumulative regret minimization problem.

Further, we adopt *random effect bandits* (Zhu & Kveton, 2022a) to learn the prior information. From the random effect bandits, we obtain the posterior distribution of the arm means, then apply the UCB-based strategy for Bayesian BAI. The algorithm works as follows. In round $t \in [n]$, it pulls the arm with the highest UCB, observes its reward, and then updates the estimated arm means and their high-probability confidence intervals. We call it *Random effect UCB Exploration (RUE)*.

We make several contributions. First, we show that instance-dependence can compromise the optimality of the `UCBE` algorithm, which can be considered as an artifact in many BAI problems. Second, we present an alternative formulation of the BAI problem that incorporates the prior distribution of arm means. Third, we bound the gap between the maximal arm and the others in probability. This result provides a principled basis for Bayesian BAI. Fourth, we propose the efficient, practical, and instance-independent UCB exploration for the BAI problem, the `RUE` algorithm. Learning the prior information, `RUE` yields superior best-arm identification performance compared to state-of-the-art methods in empirical studies. Fifth, we analyze the failure probability and simple Bayes regret of `RUE`, and derive their upper bounds of $\tilde{O}(\sqrt{K/n})$, up to logarithmic factors. Here $n$ represents the budget and $K$ denotes the number of arms. Our analysis features a sharp bound on the prior gap through order statistics, and a careful comparison of the prior gap and confidence interval for bounding the error probability. Finally, we evaluate `RUE` empirically on a range of problems and observe that it outperforms sequential halving and successive rejects in broad domains, even works better than or similarly to the infeasible `UCBE` in various domains. All proofs are in the appendix.

## 2 Exponentially Decaying Bounds in Fixed-budget BAI: An Artifact

Consider a fixed-budget BAI problem having $K$ arms with mean $\mu_k$, $k \in [K]$, and a horizon of $n$ rounds (or budgets). In round $t \in [n]$, the agent pulls arm $I_t \in [K]$ and observes its reward, drawn independently of the past. At the end of round $n$, the agent selects an arm $J_n$. The BAI problem concerns whether the final recommendation $J_n$ is the optimal one or not. For sake of simplicity, we will assume that there is a unique optimal arm. Let $i^* = \arg\max_{k \in [K]} \mu_k$ be the optimal arm and $\mu_* = \mu_{i^*}$.

Some BAI fixed-budget algorithms, such as `UCBE` and `SH`, are considered (nearly) optimal since they can achieve an exponentially decaying failure probability that depends on the instance. However the property of exponentially decaying failure probability is conditioned. To illustrate this point, we will use `UCBE` as an example. Audibert et al. (2010) defines the problem complexity of the BAI problem

$$H = \sum_{k \in [K]} \Delta_k^{-2},$$

where $\Delta_k = \mu_* - \mu_k$ for $k \neq i^*$ and $\Delta_{i^*} = \min_{k \neq i^*} \mu_* - \mu_k$ (denoted as $\Delta_{\min}$), and shows that when the exploration degree is taken appropriately, the probability of error of `UCBE` for a $K$-armed bandit with rewards in $[0, 1]$ satisfies

$$e_n \leq 2nK \exp\left[-\frac{n-K}{18H}\right].$$

However, the optimality of `UCBE` depends on $H$, which relies on $\Delta_k$, particularly the minimum gap $\Delta_{\min}$. Here, we emphasize the significant impact of the minimum gap $\Delta_{\min}$. In situations where the minimum gap

is small (i.e., $\Delta_{\min} \leq (54n^{-1}\log n)^{1/2}$, implying $H \geq 2\Delta_{\min}^{-2} = n/(27\log n)$), the upper bound has

$$2nK \exp\left[-\frac{n-K}{18H}\right] \approx 2nK \exp\left[-\frac{n}{18H}\right] \geq 2Kn^{-1/2}.$$

Unfortunately, *the small-gap condition* $\Delta_{\min} \leq (54n^{-1}\log n)^{1/2}$ may not be small in practice. For instance, in a BAI problem with $n = 10000$, the small-gap regime is defined by $\Delta_{\min} \leq 0.223$ which is not considered small by any means.

Even when considering the lower bound, the *small-gap* issue remains prevalent. In Audibert et al. (2010), it is demonstrated that for Bernoulli rewards with parameters in $[p, 1-p]$, where $p \in (0, 1/2)$, the probability of error for `UCBE` satisfies

$$e_n \geq \exp\left[-\frac{(5+o(1))n}{p(1-p)H_2}\right],$$

where $H \leq H_2 \leq \log(2K)H$ (see details in Audibert et al. (2010)). Consider an example where $p = 0.2$. In cases where $\Delta_{\min} \leq ((32n)^{-1}\log n)^{1/2}$, neglecting the contribution of the $o(1)$ term, the lower bound can be expressed as

$$\exp\left[-\frac{5n}{p(1-p)H_2}\right] \geq \exp\left(-\frac{32n}{H}\right) \geq n^{-1/2}.$$

However, it is worth noting that the small-gap condition $\Delta_{\min} \leq ((32n)^{-1}\log n)^{1/2}$ may not be small in practical scenarios. For instance, in a fixed-budget BAI problem with $n = 1000$, the small-gap regime is defined by $\Delta_{\min} \leq 0.0147$ which may not be considered very small in many fixed-budget BAI problems either.

Therefore, the exponentially decaying bounds on failure probability can be regarded as an artifact in many fixed-budget BAI problems.

At last, the presence of the *small-gap* problem also affects the choice of exploration degree, as the upper bound of `UCBE` necessitates the parameter to be less than $25(n-K)/(36H)$. However, in these scenarios, this bound is on the order $\log n$, which leads to logarithmic exploration instead of linear exploration in the fixed-budget BAI problem.

## 2.1 Scenario with Full-Information

Now, let us consider a two-arm bandit problem and examine a scenario where we have complete information: each arm $k = 1, 2$ is pulled $n$ times and its outcomes are observed. We assume that $k = 1$ is the optimal arm. In this instance, we can assume that the mean reward of each arm $k$ follows a normal distribution: $\bar{\mu}_k \sim \mathcal{N}(\mu_k, n^{-1}\sigma^2)$, where $\sigma^2$ represents the variance of reward noise. The probability of error can be bounded as follows:

$$e_n^* := \Pr(\bar{\mu}_1 \leq \bar{\mu}_2) = \Pr(\delta \geq \Delta), \tag{1}$$

where $\Delta = \mu_1 - \mu_2$ and $\delta = (\bar{\mu}_2 - \mu_2) - (\bar{\mu}_1 - \mu_1) \sim \mathcal{N}(0, 2\sigma^2/n)$. From Section 7.1 of Feller (1968), we have a lower bound of $e_n^*$:

$$e_n^* \geq \left[\left(\frac{n\Delta^2}{2\sigma^2}\right)^{-1/2} - \left(\frac{n\Delta^2}{2\sigma^2}\right)^{-3/2}\right] \frac{1}{\sqrt{2\pi}} \exp\left(-\frac{n\Delta^2}{4\sigma^2}\right) =: \bar{e}_n^*. \tag{2}$$

Now we show that this lower error bound (2) also encounters the small-gap issue. Specifically, for the small-gap condition where $\Delta \leq (2\alpha n^{-1}\log n)^{1/2}$ with $\alpha$ controlling the gap size, the error bound in the full-information scenario is given by

$$\bar{e}_n^* = (2\pi)^{-1/2}[(\sigma^{-2}\alpha\log n)^{-1/2} - (\sigma^{-2}\alpha\log n)^{-3/2}]n^{-\alpha/(2\sigma^2)}.$$

Assuming that $n$ satisfies $\sigma^2\alpha\log n > 1$, we have that when $\alpha \geq \sigma^2$,

$$\bar{e}_n^* \geq (2\pi)^{-1/2}[1/\sqrt{\log n} - 1/(\log n)^{3/2}]n^{-1/2} = \tilde{O}(n^{-1/2}).$$

Therefore, even within the realm of infeasible full information in fixed-budget BAI problems, the exponentially decaying bounds on failure probability are also artifacts. Furthermore, the error order can exceed $\tilde{O}(n^{-1/2})$ depending on the gap size.

This scenario shows that achieving exponentially decaying regrets in fixed-budget BAI is infeasible when the small-gap issue is present. Therefore, the aim of this paper is not to develop an algorithm with exponentially decaying bounds, but rather to create an algorithm that significantly addresses the small-gap issue and achieves polynomially decaying regret bounds.

# 3    A Bayesian Formulation for Best-Arm Identification

In this paper we assume that the reward, denoted as $r_k$, associated with arm $k$, is generated from an (unknown) distribution with a mean $\mu_k$. We assume that the reward noise, represented as $r_k - \mu_k$, adheres to a $\nu^2$-sub-Gaussian for a constant $\nu > 0$. We introduce the assumption of random arm means on the BAI problem. Specifically, we assume that the mean arm reward $\mu_k$ of each arm $k \in [K]$ follows the following model

$$\mu_k = \mu_0 + \delta_k \,, \tag{3}$$

where $\delta_k \sim \mathcal{N}(0, \sigma_0^2)$ and $\mathcal{N}(0, \sigma_0^2)$ is a Gaussian distribution with zero mean and variance $\sigma_0^2$. As a result, the mean reward of arm $k$, $\mu_k$, is a stochastic variable with mean $\mu_0$ and variance $\sigma_0^2$. Recently, Komiyama et al. (2023) considers a Baysian BAI setting where they assume the uniform continuity of the conditional probability density functions. Different from theirs, we make a parametric perspective on priors $\mu_k$. Our model setting has $\sigma_0^2$ to represent the variability of the arm means. With a lower variance $\sigma_0^2$, the differences among the arms are smaller. Therefore, it is harder to learn the optimal arm, as the variability of the arm means is smaller. The priors $\mu_k$, $k \in [K]$, are taken into account through $(\mu_0, \sigma_0^2)$.

Different from the algorithms that rely on $H$, in the Bayesian BAI setting $\mu_* - \mu_k$ for $k \neq i^*$, can be arbitrarily small. Fortunately we can control the probability that the gap $\mu_* - \mu_k$ is less than $\alpha$ for any $\alpha > 0$. This is our key in the Bayesian BAI problem. Define

$$e_*(\alpha) = \Pr\left(\sup_{k \neq i^*} \mu_* - \mu_k \leq \alpha\right),$$

for any $\alpha > 0$. The probability $e_*$ represents the likelihood that the optimal arm $i^*$ is at least $\alpha$ better than the other arms. In other words, it reflects the probability of obtaining a gap between $i^*$ and the other arms that is less than $\alpha$ based on the prior distribution. In the BAI problem, $\alpha$ decreases as the numbers of pulls increases, allowing for control over the probability. This highlights the inherent difficulty of the Bayesian BAI problem when dealing with the prior distribution of $(\mu_k)_{k \in [K]}$.

**Theorem 3.1.** *Assume $\mu_k$, for $k \in [K]$, are independently and identically distributed from $\mathcal{N}(\mu_0, \sigma_0^2)$. Then for $\alpha > 0$,*

$$e_*(\alpha) \leq c_K \alpha / \sigma_0.$$

*where $c_K = 4\sqrt{2}\ln K \sqrt{\ln\left(\frac{K}{4\sqrt{2\pi}\ln K}\right)} + \frac{2}{\sqrt{2\pi}}$.*

This theorem provides a principled basis for Bayesian BAI in the following aspect. For any $\alpha > 0$, the probability of bounding the gap by $\alpha$ is inversely proportional to the s.d. of arm means, $\sigma_0$, and is logarithmic of the number of arms, $K$. It means that the effect of increasing $K$ on $e_*$ is negligible up to logarithmic factor.

We end this section by comparing $\sigma_0^2$ with $\Delta_{\min}$. In our Bayesian BAI setting, $\sigma_0^2 = \mathbb{E}[(\mu_k - \mu_0)^2]$, which represents the expected value of the squared deviation of $\mu_k$ from $\mu_0$. This measure does not rely on the minimum gap $\Delta_{\min}$. In other words, $\sigma_0^2$ can be $O(1)$ even if $\Delta_{\min} = o(1)$. Typically, $\sigma_0^2 = O(1)$, which largely avoids the *small-gap* problem. Furthermore, as shown in Section 5, our analysis accommodates a smaller $\sigma_0^2 = O(1/K)$.

## 4 Algorithm

In Section 4.1 we show the Bayesian estimation, and provide a heuristic motivation for why the use of confidence intervals is applicable to Bayesian BAI. At last we propose a variant of the UCB algorithm in Section 4.2.

### 4.1 Estimation

For arm $k$ and round $t$, we denote by $T_{k,t}$ the number of its pulls by round $t$, and by $r_{k,1}, \ldots, r_{k,T_{k,t}}$ the sequence of its associated rewards.

We use Gaussian likelihood function to design our algorithm. More precisely, suppose that the likelihood of reward $r_{k,T_{k,t}}$ at time $t$, given $\mu_k$, were given by the pdf of Gaussian distribution $\mathcal{N}(\mu_k, \sigma^2)$, where we take $\sigma^2 = \delta^{-1}\nu^2$ for $0 < \delta \leq 1$. We emphasize that the Gaussian likelihood model for rewards is only used above to design the algorithm. The assumptions on the actual reward distribution are the $\nu^2$-sub-Gaussian assumption. This setup is analogous to the one described in Agrawal & Goyal (2013).

Let the history $H_t = (I_\ell, r_{I_\ell, T_{I_\ell, \ell}})_{\ell=1}^{t-1}$. In the context where the prior for $\mu_k$ is given by $\mathcal{N}(\mu_0, \sigma_0^2)$, deriving the posterior distribution utilized by our algorithm is straightforward:

$$\mu_k | H_t \sim \mathcal{N}(\hat{\mu}_{k,t}, \tau_{k,t}^2). \tag{4}$$

Here, the posterior mean $\hat{\mu}_{k,t}$ of $\mu_k$ is given by

$$\hat{\mu}_{k,t} = (1 - w_{k,t})\bar{r}_{0,t} + w_{k,t}\bar{r}_{k,t}, \tag{5}$$

where $w_{k,t} = \sigma_0^2 / (\sigma_0^2 + T_{k,t}^{-1}\sigma^2)$ and

$$\bar{r}_{0,t} = \left[ \sum_{k=1}^{K} (1 - w_{k,t}) T_{k,t} \right]^{-1} \sum_{k=1}^{K} (1 - w_{k,t}) \sum_{j=1}^{T_{k,t}} r_{k,j}.$$

The posterior variance $\tau_{k,t}^2$ is given by

$$\tau_{k,t}^2 = \frac{w_{k,t}\sigma^2}{T_{k,t}} + \frac{(1 - w_{k,t})^2 \sigma^2}{\sum\limits_{k=1}^{K} T_{k,t}(1 - w_{k,t})}, . \tag{6}$$

The following proposition motivates the use of confidence intervals in Bayesian BAI.

**Proposition 4.1.** *Let* $\Delta_k = \mu_{i^*} - \mu_k$. *For any sub-optimal arm* $k \neq i^*$, *we have*

$$Pr(\hat{\mu}_{k,n} \geq \hat{\mu}_{i^*,n} | \Delta_k, H_n) \leq \exp\left[ -\frac{\Delta_k^2}{8\tau_{k,n}^2} \right] + \exp\left[ -\frac{\Delta_k^2}{8\tau_{i^*,n}^2} \right].$$

Proposition 4.1 shows that the probability of failing to identify the best arm depends on the gap $\Delta_k$ and $\tau_{k,n}^2$ for $k \in [K]$. This result shows that the widths of the confidence intervals affect the failure probability. Motivated by this observation, we design an efficient UCB-based BAI algorithm by using the estimates from random effect bandits.

### 4.2 Random-Effect UCB Exploration

We apply random effect bandits to BAI, and propose a novel UCB-based exploration algorithm, called *Random effect UCB Exploration (*`RUE`*)*. In `RUE`, the upper confidence bound of arm $k$ in round $t$ is

$$U_{k,t} = \hat{\mu}_{k,t-1} + \sqrt{2\tau_{k,t-1}^2 \log n}.$$

Different from `ReUCB` (Zhu & Kveton, 2022a), which minimizes the cumulative regret and uses the degree of exploration $2 \log t$, `RUE` uses the degree of exploration $2 \log n$, since BAI requires high-probability confidence intervals only at the final round. In round $t$, the algorithm pulls the arm with the highest UCB $I_t = \arg\max_{k \in [K]} U_{k,t}$ and collects the associated reward. Any fixed tie-breaking rule can be used for multiple maximal.

---

**Algorithm 1** `RUE` for best-arm identification.

---

1: Initialization: Pull each arm twice
2: **for** $t = 2K+1, \ldots, n$ **do**
3:      Calculate $U_{k,t} = \hat\mu_{k,t-1} + \sqrt{2\tau_{k,t-1}^2 \log n}$
4:      Pull the arm with the highest $U_{k,t}$ for $k \in [K]$
5:      Collect the reward associated the chosen arm
6: **end for**
7: Return estimated best arm $J_n = \arg\max_{k \in [K]} \hat\mu_{k,n}$

---

In `RUE`, the priors $(\mu_k)_{k \in [K]}$ are taken into account through the variances $\sigma_0^2$ and $\sigma^2$. Various methods for obtaining consistent estimators of $\hat\sigma_0^2$ and $\hat\sigma^2$ are available, including the method of moments, maximum likelihood, and restricted maximum likelihood. See Robinson (1991) for details. One practical implication of this is that unlike `UCBE`, our algorithm focuses on learning the prior to implement the algorithm. This feature of `RUE` can have surprising practical benefits.

## 5 Analysis

We first bound the probability that `RUE` fails to identify the best arm. Let $e_n$ be the probability that `RUE` fails to identify the best arm

$$e_n = \Pr(J_n \neq i^*),$$

which is over both the stochastic rewards and randomness in arm means $(\mu_k)_{k \in [K]}$. The main novelty in our analysis is that we control the failure probability of carefully comparing the gap of $(\mu_k)_{k \in [K]}$ and the confidence bounds.

**Theorem 5.1.** *Consider Algorithm 1 in a $K$-armed bandit with a budget $n \geq 4(K-1)$. Denote $\rho = \sqrt{(K(\sigma_0^2 + \sigma^2) + \sigma_0^{-2}\sigma^2)/(K(\sigma_0^2 + \sigma^2) + \sigma_0^2)}$ and $H_b = (K + \sigma_0^{-2}\sigma^2)\sigma^2$. Then the failure probability of Algorithm 1 is*

$$e_n \leq \gamma \sqrt{\frac{H_b(K-1)\log n}{nK}} + \gamma \sqrt{\frac{H_b \log n}{K(n - 4(K-1))}} + 2Kn^{-mK+1} + 2Kn^{-\frac{\sigma^2 m}{\delta(2\sigma_0^2 + \sigma^2)} + 1}.$$

*where $\gamma = 2(1 + 4\rho)^{-1}(2 + 4\rho)c_K$ and $m = (1 + K^{-1}\sigma_0^2/(\sigma_0^2 + \sigma^2))(1 + 4\rho)^{-2}\sigma^{-2}\sigma_0^2$.*

Following the Bayesian failure probability of `RUE` in Theorem 5.1, we bound its simple Bayes regret

$$\mathrm{sr}_n = \mathbb{E}[\mu_* - \mu_{J_n}],$$

where the expectation is over stochastic rewards and the randomness in $(\mu_k)_{k \in [K]}$. By applying Theorem 5.1, we demonstrate in the following theorem that the simple Bayes regret is $\tilde O(\sqrt{K/n})$.

**Theorem 5.2.** *Consider Algorithm 1 in a $K$-armed bandit with a budget $n > 4(K-1)$. Under the condition of Theorem 5.1, the simple Bayes regret of `RUE` is*

$$\mathrm{sr}_n \leq \kappa \sqrt{\frac{2H_b(K-1)\log n}{nK}} + \kappa \sqrt{\frac{2H_b \log n}{K(n - 4(K-1))}} + 4\sigma_0\sqrt{2\log K}\left(Kn^{-mK+1} + Kn^{-\frac{\sigma^2 m}{\delta(2\sigma_0^2 + \sigma^2)} + 1}\right).$$

*where $\kappa = 4(1 + 4\rho)^{-1}(2 + 4\rho)c_K\sqrt{\log K}$.*

## 5.1 Discussion

The parameter $\delta$ in Theorems 5.1 and 5.2 controls the last term of the bounds. When $\delta \leq \frac{\sigma^2 m}{2(2\sigma_0^2+\sigma^2)}$, the last term is $\tilde{O}(K/n)$. As introduced in Section 4.1, $\delta = \sigma^{-2}\nu^2$ denotes the ratio of $\nu^2$ to the variance $\sigma^2$ of the Gaussian likelihood designed in the algorithm. The condition on $\delta$ implies that a larger $\sigma^2$ than $\nu^2$ is required in the Gaussian likelihood. This requirement is analogous to that in Agrawal & Goyal (2013). Moreover, $Kn^{-mK+1} = O(K/n)$ when $mK \geq 2$. Typically when $K \gg \sigma_0^2, \sigma^2$, we have $\rho \approx 1$, which impllies $m \approx \sigma_0^2 \sigma^{-2}/25$. Clearly, the condition $mk \geq 2$ permits $\sigma_0^2 = O(1/K)$. Hence, Theorems 5.1 and 5.2 state that the upper bounds on the failure probability and the simple Bayes regret are $\tilde{O}(\sqrt{H_b/n})$ respectively.

Here we characterize the hardness of the task using the quantity $H_b = (K + \sigma_0^{-2}\sigma^2)\sigma^2$, which relies on the parameter set of bandits $(K, \sigma_0^2, \sigma^2)$. The quantity $H_b$ increases as the number of arms $K$ increases, the noise variance $\sigma^2$ increases, or the variability of the arms' means $\sigma^2$ decreases. Obviously, $H_b = O(K)$ when $\sigma_0^2 = O(1)$ or $\sigma_0^2 = O(1/K)$, given that $\sigma^2 = O(1)$. In this case, the upper bounds are $\tilde{O}(\sqrt{K/n})$. This indicates that the bounds remain $\tilde{O}(\sqrt{K/n})$, even when $\sigma_0^2$ are small (i.e., $\sigma_0^2 = O(1/K)$).

As demonstrated in Section 2, the upper bound on the failure probability of `UCBE` exhibits an exponential decay that is conditioned on the value of $\Delta_{\min}$. When $\Delta_{\min}$ is small, their bound degenerates to a polynomially decay, resulting in a failure probability of $\tilde{O}(Kn^{-\eta})$, where $\eta > 0$ depends on how small $\Delta_{\min}$ is. In contrast, the bounds provided in Theorems 5.1 and 5.2 are instance-independent, meaning that they solely depend on the budget $n$ and the bandit class defined by the number of arms and the variances, denoted as $(K, \sigma_0^2, \sigma^2)$, for which `RUE` is designed. These bounds are not influenced by the specific instances within the class, ensuring their independence from the particular characteristics of each instance. Nevertheless, as shown in Section 3, $\sigma_0^2$ characterizes the variability of the arms' means $\mu_k$ in relation to $\mu_0$, thereby permitting very small $\Delta_{\min}$. Consequently, the *small-gap* issue does not pose a significant challenge in our algorithm.

## 5.2 Proof Outline

Here we outline the proof. Comprehensive details can be found in the Appendix. Without loss of generality, we assume arm 1 is the optimal arm, i.e., $\mu_{i^*} = \mu_1$. Note that the initial round is $2K + 1$, since every arm is pulled twice in the first $2K$ rounds. Denote

$$c_{k,t-1} = \sqrt{2\tau_{k,t-1}^2 \log n}.$$

We define the events that all confidence intervals from round $2K + 1$ to round $n$ hold as,

$$\mathcal{E} = \{\forall k \in [K], t \in \{2K + 1, \ldots, n\} : |\mu_k - \hat{\mu}_{k,t}| \leq \eta c_{k,t}\},$$

where $\eta = 1/(1 + 4\rho)$.

The error probability is decomposed as

$$\begin{aligned}
&\Pr\left(\hat{\mu}_{J_n,n} - \hat{\mu}_{1,n} > 0\right) \\
=&\Pr\left((\hat{\mu}_{J_n,n} - \mu_{J_n}) - (\hat{\mu}_{1,n} - \mu_1) > \Delta_{J_n} | \mathcal{E}\right)\Pr(\mathcal{E}) + \Pr\left((\hat{\mu}_{J_n,n} - \mu_{J_n}) - (\hat{\mu}_{1,n} - \mu_1) > \Delta_{J_n} | \bar{\mathcal{E}}\right)\Pr(\bar{\mathcal{E}}) \\
\leq&\Pr\left(\Delta_{J_n} < \eta(c_{1,n} + c_{J_n,n})|\mathcal{E}\right) + \Pr(\bar{\mathcal{E}}),
\end{aligned} \tag{7}$$

From (7), $e_n$ is decomposed into two terms. The first term is to compare the prior's gap with the upper confidence bounds. The second term is the probability that the confidence intervals do not hold.

Denote $\beta = 1 + K^{-1}\sigma_0^{-2}\sigma^2$ and

$$\tilde{c}_{k,n} = \sqrt{\frac{2\sigma_0^2 \sigma^2 \log n}{T_{k,n}\sigma_0^2 + \sigma^2}}, \forall k \in [K]. \tag{8}$$

Because $\tilde{c}_{k,n}$ can be bounded by $c_{k,n}$: $c_{k,n} \leq \sqrt{\beta}\tilde{c}_{k,n}$ as shown in Zhu & Kveton (2022a). Thus, we have that

$$\Pr\left(\Delta_{J_n} < \eta(c_{1,n} + c_{J_n,n})\right) \leq \Pr\left(\Delta_{J_n} < \eta\sqrt{\beta}(\tilde{c}_{1,n} + \tilde{c}_{J_n,n})\right).$$

Now we investigate to bound

$$\Pr\left(\Delta_k \leq \eta\sqrt{\beta}(\tilde{c}_{1,n} + \tilde{c}_{k,n})|\mathcal{E}\right)$$

for any $k \neq 1$. Denote $\Delta_{\min} = \min_{k \neq 1} \Delta_k$. We define the following event of comparing $\tilde{c}_{1,n}$ with $\Delta_{\min}$:

$$\mathcal{E}_1 := \{\tilde{c}_{1,n} \leq \Delta_{\min}/(\sqrt{\beta}(1+\eta))\}.$$

We can show that

$$\Pr\left(\Delta_k < \eta\sqrt{\beta}(\tilde{c}_{1,n} + \tilde{c}_{k,n})|\mathcal{E}\right)$$

$$\leq \Pr\left(\bar{\mathcal{E}}_1|\mathcal{E}\right) = \Pr\left(\Delta_{\min} < (1+\eta)\sqrt{\frac{\beta\sigma_0^2\sigma^2\log n}{T_{1,n}\sigma_0^2 + \sigma^2}}|\mathcal{E}\right).$$

Then we investigate $T_{1,n}$. We show that

$$T_{1,n} \geq n - 2(K-1)(1+\eta)^2\Delta_{\min}^{-2}\beta\sigma^2\log n - 2(K-1). \tag{9}$$

For decoupling $T_{1,n}$ and $\Delta_{\min}$, we define the following event of controlling the minimum gap:

$$\mathcal{E}_2 := \left\{\Delta_{\min} \geq 2(1+\eta)\sqrt{(K-1)\beta\sigma^2 n^{-1}\log n}\right\}.$$

On the event $\mathcal{E}_2$, (9) follows that

$$\sigma_0^2 T_{1,n} + \sigma^2 \geq \sigma_0^2 n/2 - 2\sigma_0^2(K-1) + \sigma^2.$$

Thus, we have that

$$\Pr\left(\Delta_{\min} < (1+\eta)\sqrt{\frac{2\beta\sigma_0^2\sigma^2\log n}{T_{1,n}\sigma_0^2 + \sigma^2}}|\mathcal{E}\right)$$

$$\leq \Pr\left(\Delta_{\min} < (1+\eta)\sqrt{\frac{2\beta\sigma_0^2\sigma^2\log n}{\sigma_0^2 n/2 - 2\sigma_0^2(K-1) + \sigma^2}}|\mathcal{E}\right) + \Pr\left(\bar{\mathcal{E}}_2|\mathcal{E}\right)$$

$$\leq \gamma\left(\sqrt{\frac{\beta\sigma^2\log n}{\sigma_0^2 n - 4\sigma_0^2(K-1) + 2\sigma^2}} + \sqrt{\frac{\beta\sigma^2\log n}{\sigma_0^2 n/(K-1)}}\right),$$

where the last step is from applying Theorem 3.1. Therefore, the first term of the decomposition in (7) is bounded.

At last, we investigate the second term of the decomposition in (7). Given our analysis of the algorithm, it is imperative to note that these models may be entirely unrelated to the actual reward distribution. Consequently, we cannot make the assumption that the posterior distribution of $\mu_k$—given the historical data—is Gaussian, with the posterior mean $\hat{\mu}_{k,t}$. Instead, we opt for a decomposition approach, as detailed below. Denote $\beta_1 = 1 + K^{-1}\sigma_0^2/(\sigma_0^2 + \sigma^2)$. Define the following event:

$$\mathcal{E}_{3k} := \left\{|\bar{r}_{k,t} - \mu_k| \leq \eta\sqrt{\beta_1}\tilde{c}_{k,t}/2\right\}.$$

We have that

$$\Pr\left(\bar{\mathcal{E}}\right) \leq \sum_{k=1}^{K}\sum_{t=1}^{n}\Pr\left(\frac{\sigma^2|\bar{r}_{0,t} - \mu_k|}{T_{k,t}\sigma_0^2 + \sigma^2} > \frac{\eta\tilde{c}_{k,t}}{2}\right) + \sum_{k=1}^{K}\sum_{t=1}^{n}\Pr\left(\bar{\mathcal{E}}_{3k}\right). \tag{10}$$

We aim to bound the two terms in (10). Utilizing the properties of sub-Gaussian distributions, it follows that $\bar{r}_{0,t} - \mu_0$ is sub-Gaussian. By applying the sub-Gaussian tail inequality, we obtain that

$$\Pr\left(\frac{\sigma^2|\bar{r}_{0,t} - \mu_k|}{T_{k,t}\sigma_0^2 + \sigma^2} > \frac{\eta\tilde{c}_{k,t}}{2}\right) \leq \exp\left(-\frac{\beta_1\sigma_0^2 K\log n}{(1+4\rho)^2\sigma^2}\right).$$

On the other hand, exploiting the sub-Gaussian property of $\bar{r}_{k,t} - \mu_k$, we have

$$\Pr\left(\bar{\mathcal{E}}_{3k}\right) \leq \exp\left(-\frac{\beta_1 \sigma_0^2 \log n}{\delta(1+4\rho)^2(2\sigma_0^2 + \sigma^2)}\right).$$

Therefore, the theorem is proved.

## 6 Experiments

We conduct two kinds of experiments. In Section 6.1, $\mu_k$ are random, where various settings are considered. In Section 6.2, $\mu_k$ are fixed. Note that our modeling assumptions are violated here. We show these experiments of fixed $\mu_k$ because they are benchmarks established by Audibert et al. (2010) and Karnin et al. (2013).

Our baselines include the state-of-the-art Successive Rejects (SR) (Audibert et al., 2010), Sequential Halving (SH) (Karnin et al., 2013), the UCB-exploration (UCBE) (Audibert et al., 2010), and Top-Two Thompson sampling (TTTS) (Russo, 2020). As mentioned in Section 4.2, in the implementation of RUE, we use the estimates of the variances $\sigma_0^2$ and $\sigma^2$. Therefore, no hyperparameters are required for RUE. UCBE is implemented with parameter $a = 2n/H$, since this parameter works the best overall according to Audibert et al. (2010) and Karnin et al. (2013). We do not report the adaptive variant of UCBE because it performs much worse than UCBE; even worse than SH (Karnin et al., 2013). Note UCBE is infeasible since it requires the knowledge of a problem complexity parameter $H$, which depends on gaps. Additionally, we assess the two-stage algorithm proposed by Komiyama et al. (2023), but it does not perform well in our experiments (see Figure 3 in the Appendix). Consequently, we exclude their method from our comparison.

### 6.1 Random $\mu_k$

For random $\mu_k$, we have the following three setups:
(R1) Gaussian rewards with mean $\mu_k$ and variance $\sigma^2 = 1$, where $\mu_k \sim \mathcal{U}(0, 0.5)$ for $k \in [K]$.
(R2) The same $\mu_k$ as R1, but the rewards are Bernoulli rewards with means $\mu_k$.
(R3) Bernoulli rewards with $\mu_k \sim \mathcal{U}(0, 0.5)$ for $k \in [K]$.
These setups allow us to explore how the performance of RUE compare against benchmarks under various distributions of noise and reward means. Although we assume a Gaussian distribution in our Bayesian BAI formulation, we also evaluate the performance of RUE for R3 when the assumption does not hold.

We report the performance for $K = 20$. Due to the randomness of $\mu_k$, the gaps and the difficulty of BAI varies. Therefore, we conduct our experiments on 50 sampled $\mu_1, \ldots, \mu_K$. For each set, we evaluate the performance of RUE and state-of-the-art algorithms, and then report the average performance. Like the case of fixed $\mu_k$, we also set $a = 2$ through three random setups.

The maximum budgets are set to $N = 5000$ for R1 and R2, and to $N = 12000$ for R3. We choose these values based on the median complexity terms in our experiments, which are $H \approx 2000$ for R1 and R2, and $H \approx 5500$ for R3. Then we study various budget settings $n \in \{N/4, N/2, N\}$, so that we can show how the performance varies when the budget is less than $H$ and double of $H$.

In Figure 1, we report the average performance over 50 experiments, and also boxplots of the relative performance of the baselines SR, SH, UCBE, and TTTS with respect to RUE. We observe that, except for performing better or worse than TTTS in R3, RUE dominates other methods in all three setups, and even works better than UCBE. For example, for the case of $n = N/2$ in R3, the error probabilities of RUE are smaller 23%, 24%, and 31%, respectively, than UCBE, SH, and SR. These results shows the flexibility of RUE across various distributions of reward noise and across various distributions of reward means.

### 6.2 Fixed $\mu_k$

Like Karnin et al. (2013), we study six different experimental setups to comprehensively assess the RUE's performance:
(F1) One group of suboptimal arms: $\mu_k = 0.45$ for $k \geq 2$.

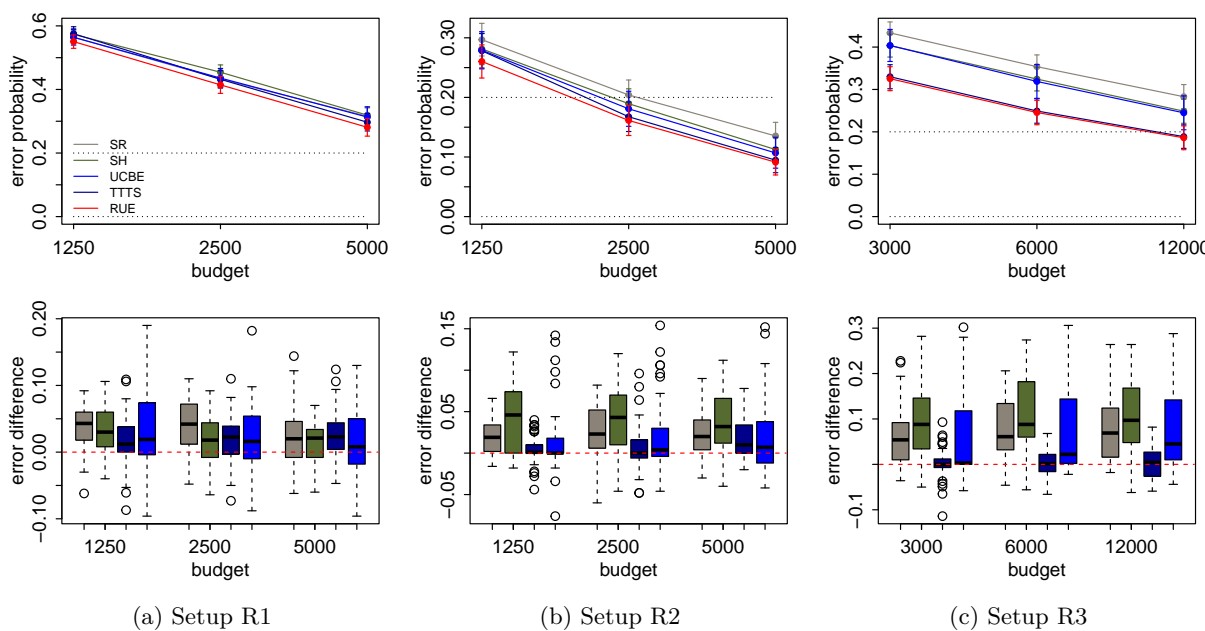

(a) Setup R1          (b) Setup R2          (c) Setup R3

Figure 1: Random $\mu_k$. Upper panel: the average performance among 50 experiments. The bars denote the standard error of the mean among 50 experiments. The lower panel: the error difference of the performance of the baselines, SR, SH, and UCBE, respectively, with respect to that of RUE among all 50 experiments.

(F2) Two groups of suboptimal arms: $\mu_k = 0.45$ for $k = 2, \ldots, 8$ and $\mu_k = 0.3$ otherwise.

(F3) Three groups of suboptimal arms: $\mu_k = 0.48$ for $k = 2, \ldots, 5$, $\mu_k = 0.4$ for $k = 6, \ldots, 13$ and $\mu_k = 0.3$ otherwise.

(F4) Arithmetic: The suboptimality of the arms form an arithmetic series where $\mu_2 = 0.5 - 1/(5K)$ and $\mu_K = 0.25$.

(F5) Geometic: The suboptimality of the arms form an geometric series where $\mu_2 = 0.5 - 1/(5K)$ and $\mu_K = 0.25$.

(F6) One real competitor: $\mu_2 = 0.5 - 1/(10K)$ and $\mu_k = 0.45$ for $k = 3, \ldots, K$.

In all setups, the reward distributions are Bernoulli and the mean reward of the best arm is 0.5. The number of arms is $K = 20$. We also examine $K \in \{40, 80\}$ in Figure 4 of Appendix, to show how RUE scales with $K$.

We set $2\lceil H \rceil$ as the maximal budget for matching the hardness and for the limit of resources. Then we study various budget settings $n \in \{\lceil H/2 \rceil, \lceil H \rceil, 2\lceil H \rceil\}$, so that we can show the performance when the budget is less or more than $H$. In RUE, we plug in the estimators of the variances $\sigma^2$ and $\sigma_0^2$ as in Zhu & Kveton (2022a).

Figure 2 shows results for our six problems. We have the following observations. First, RUE consistently outperforms SH and SR in all problems (except for the $n = \lceil H/2 \rceil, \lceil H \rceil$ of F1, where it's a little worse than SR). Take F2 as an example. For the budget $\lceil H/2 \rceil, \lceil H \rceil$, and $2\lceil H \rceil\}$, the error probabilities of RUE are smaller 10%, 20%, and 66%, respectively, than SH. Second, comparing to the infeasible UCBE, RUE outperforms it in F3,F5-F6, performs similarly to it in F4, and performs worse than it in F1 and F2. Third, comparing to TTTS, RUE outperforms it in most cases but performs slightly worse in $n = 2\lceil H \rceil\}$ for F4 and F6. Fourth, comparing with various $K \in \{40, 80\}$ in Figure 4 of Appendix, we observe that the outperformance of RUE over others grows as $K$ increases. In summary, the observations suggest that RUE is expected to work well in various domains.

# 7 Related Work

Bubeck et al. (2009) showed that algorithms with at most logarithmic cumulative regret, such as UCB1 (Auer et al., 2002), are not suitable for BAI; and proposed to explore more aggressively using $O(\sqrt{n})$ confidence

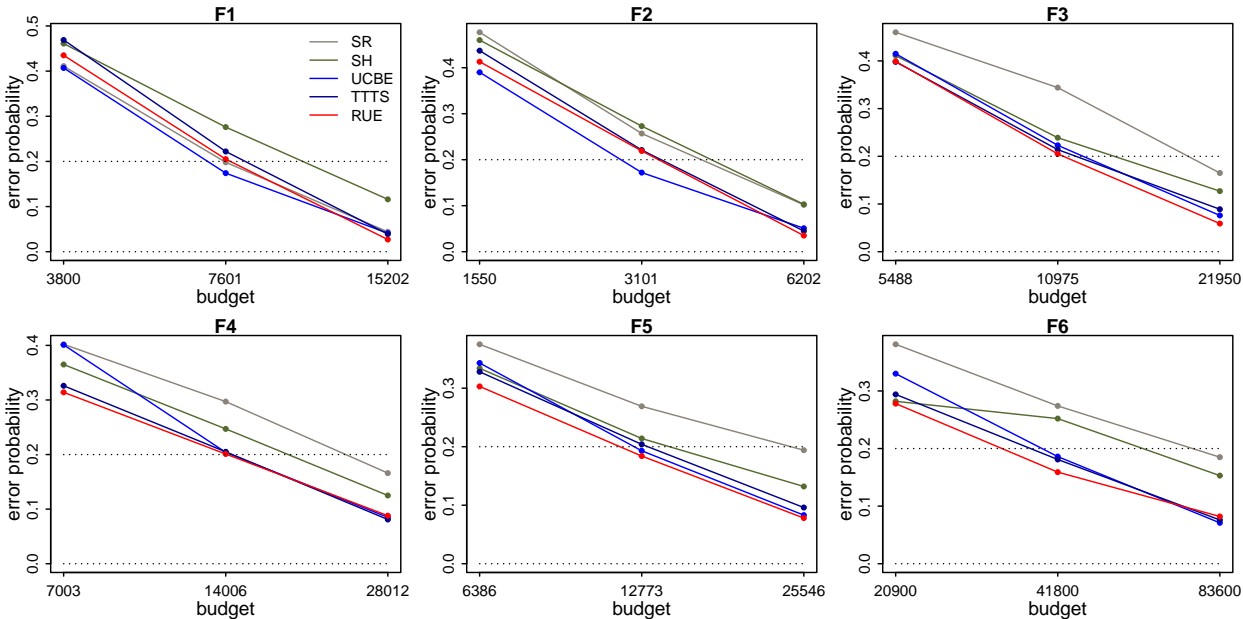

Figure 2: Fixed $\mu_k$. All standard errors are less than 0.016 and not reported. Each experiment is with budgets $H/2$, $H$, and $2H$ (labels of the x axis). The dotted lines denote error probabilities 0 and 0.2, just for visual clarity.

intervals. Motivated by it, Audibert et al. (2010) considered the fixed budget setting, where `UCBE` and successive rejects are proposed for BAI. `UCBE` and its adaptive version have $O(\sqrt{n})$ confidence intervals. In comparison, our work shows that a UCB-based algorithm with $O(\sqrt{\log n})$ confidence intervals performs well in BAI. On the other hand, the infeasible `UCBE` algorithm depends on the unknown gap, while the estimated gap makes the adaptive `UCBE` less efficient. Our algorithm does not rely on the actual or estimated gaps. Karnin et al. (2013) proposed sequential halving, which is popular in hyperparameter optimization (Jamieson & Talwalkar, 2015; Li et al., 2018). Different from the method, this work focuses on efficient exploration based on upper confidence bounds.

Fixed-confidence setting was introduced by Even-Dar et al. (2006), who proposed successive elimination for BAI. Mannor & Tsitsiklis (2004) derived tight distribution-dependent lower bounds for several variants of successive elimination, Jamieson et al. (2014) proposed lil-UCB; Tánczos et al. (2017) extended lil-UCB to the KL-based confidence bounds (Garivier & Cappe, 2001; Kaufmann & Kalyanakrishnan, 2013), and Shang et al. (2020) adjusted TTTS (Russo, 2020) for fixed-confidence guarantees. In comparison, we focus on the fixed-budget setting.

Although the fixed-confidence setting and the fixed-budget setting seem "dual" to each other, they perform differently in several domains. Recently, Qin (2022) proposed an open problem regarding whether there exists an algorithm other than uniform sampling itself that performs uniformly no worse than uniform sampling in the fixed-budget setting. Degenne (2023) and Wang et al. (2023) demonstrated that in the fixed-budget setting, there are no such universally superior adaptive algorithms in several BAI problems. These studies show that the expected probability of error of BAI in the fixed-budget setting may be significantly different from that in the fixed-confidence setting. Our paper presents another observation in the fixed-budget setting: exponentially decaying bounds in fixed-budget BAI are an artifact.

Zhu & Kveton (2022a;b) proposed random effect bandits for cumulative regret minimization. Our work can be viewed as an extension of Zhu & Kveton (2022a) to best-arm identification. We show that the prior information is helpful to develop an efficient, practical UCB exploration algorithm for the Bayesian BAI problem.

Recently, Bayesian BAI has received attention. Russo (2020) proposed Bayesian algorithms, top-two variants of Thompson sampling (TTTS), that are tailored to identifying the best arm. His analysis focused on the frequentist consistency and rate of convergence of the posterior distribution. Shang et al. (2020) followed his work and proposed a variant of TTTS for justifying its use for fixed-confidence guarantees. Different from theirs, we propose a variant of UCB exploration by using the prior information and show its efficiency by analyzing the error probability and the simple Bayes regret. Komiyama et al. (2023) and Azizi et al. (2023) derived a lower bound for this setting and a two-phase algorithm that matches it. However, empirically their algorithm works badly as shown in Appendix. Recently, Atsidakou et al. (2023) introduced a Bayesian version of the SH algorithm and provided prior-dependent bound on the probability of error in multi-armed bandits. Furthermore, Nguyen et al. (2024) established upper prior-dependent bounds on the expected probability of error of prior-informed BAI in Structured Bandits.

Our Bayesian BAI formulation is also related to Bayesian optimization (Snoek et al., 2012) which assumes a Gaussian process prior and updates the posterior with new observations. Similar to Bayesian optimization, the Bayesian BAI setting uses a Gaussian prior to learn configuration evaluations. However, unlike Bayesian optimization, the Gaussian prior in the Bayesian BAI setting models reward means of individual arms.

## 8 Conclusions

We introduce a formulation of the Bayesian fixed-budget BAI problem by modeling the arm means, and propose `RUE`, an efficient, instance-independent UCB exploration for fixed-budget BAI. We empirically show that `RUE` outperforms `SH` and `SR` in broad domains, even works better than or similarly to the infeasible `UCBE` in various domains. We derive $\tilde{O}(\sqrt{K/n})$ bounds on its Bayesian failure probability and simple Bayes regret. Inspired by Li et al. (2018), which demonstrates that BAI can be applied to hyper-parameter optimization, our proposed `RUE` has potential for use in hyper-parameter optimization. This application warrants serious investigation in the future.

Nevertheless, an inherent limitation of this study is the absence of a corresponding lower bound, as obtaining one for fixed-budget BAI is a challenging task (Qin, 2022). Another limitation is that the analysis in this paper only focuses on the Gaussian settings. Nevertheless, `RUE` does not assume any distributional form, since the setting in (3) does not assume any particular distribution, but only assumes that the first- and second-order moments of $\mu_k$ are bounded. Moreover, our empirical results show that `RUE` works well in broad domains where the Gaussian assumptions are violated. Therefore, an interesting question is to provide the bound on the failure probability under sub-Gaussian settings.

In our analysis, we assume that $\sigma^2$ and $\sigma_0^2$ are known. However, in practice, we substitute these parameters with their estimates. Investigating the effect of this substitution in the Bayesian setting is extremely challenging since we need to integrate the error probability over the posterior of these parameters. We leave this challenging problem as a future direction for research. Nevertheless, we expect this effect to be small since the agent's estimates of these parameters should converge to their true values as the agent gathers more data.

### Author Contributions

Rong J.B. Zhu designed and performed the research, performed the analysis and experiments, and wrote the paper. Yanqi Qiu provided the proof for Lemma A.1.

### Acknowledgements

We are very grateful to Branislav Kveton for insightful discussions and invaluable comments on the preliminary version of this paper. We thank the Assigned Action Editor, Marcello Restelli, and the three reviewers for their insightful comments and suggestions that significantly improve this paper.

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

# A Appendix

## A.1 Proof of Theorem 3.1

First we provide the following lemma, which provides a basic tool for bounding the gap $\mu_* - \mu_k$. Then we show that the probability $e_*$ depends on $\sigma_0^2$ and $\ln K$ by following Lemma A.1.

**Lemma A.1.** *Assume $X_k$ for $k = 1, \ldots, K$, are independent and identically distributed from $\mathcal{N}(0, 1)$. Denote $X_{(1)} \geq X_{(2)} \geq \cdots X_{(K)}$ the non-increasing re-ordering of $X_1, \ldots, X_K$. Then there exists a constant $C > 0$, such that for all integers $K \geq 2$ and for $\alpha > 0$,*

$$Pr(X_{(1)} - X_{(2)} \leq \alpha) \leq C(\ln K)^{3/2}\alpha.$$

*Proof.* Denote $\eta(\alpha) = \Pr(X_{(1)} - X_{(2)} \leq \alpha)$. Let $\Phi(x)$ and $f(x)$ be the cumulative distribution function and the density function, respectively, of $X_i$. From the joint distribution of $(X_{(1)}, X_{(2)})$ (Fact 2 in the Appendix), we have that

$$\eta(\alpha) = K(K-1) \int_{0 \leq x_1 - x_2 \leq \alpha} \Phi(x_2)^{K-2} f(x_2) f(x_1) dx_2 dx_1$$

$$= K(K-1) \int_{0 \leq z \leq \alpha; x_2 \in \mathbb{R}} \Phi(x_2)^{K-2} f(x_2) f(x_2 + z) dx_2 dz$$

$$= K(K-1) \int_{x_2 \in \mathbb{R}} \Phi(x_2)^{K-2} f(x_2)[\Phi(x_2 + \alpha) - \Phi(x_2)] dx_2$$

$$= K \int_{x_2 \in \mathbb{R}} \Phi(x_2 + \alpha) d\Phi(x_2)^{K-1} - (K-1) \int_{x_2 \in \mathbb{R}} d\Phi(x_2)^K$$

$$= K - K \int_{x_2 \in \mathbb{R}} \Phi(x_2)^{K-1} f(x_2 + \alpha) dx_2 - (K-1)$$

$$= 1 - K \int_{x_2 \in \mathbb{R}} \Phi(x_2)^{K-1} f(x_2 + \alpha) dx_2.$$

It follows that

$$\eta'(\alpha) = -K \int_{x_2 \in \mathbb{R}} \Phi(x_2)^{K-1} f'(x_2 + \alpha) dx_2.$$

Fix $c > 0$ (which will be choosen later on). For any integer $K \geq 2$, let $a_K > 0$ be the unique solution to the equation

$$a_K e^{a_K^2/2} = \frac{cK}{\ln K}.$$

Set

$$T_1(\alpha) = -K \int_{x_2 > a_K} \Phi(x_2)^{K-1} f'(x_2 + \alpha) dx_2;$$

$$T_2(\alpha) = -K \int_{x_2 \leq a_K} \Phi(x_2)^{K-1} f'(x_2 + \alpha) dx_2.$$

We can now write

$$\eta'(\alpha) = T_1(\alpha) + T_2(\alpha).$$

For the term $T_1(\alpha)$, by noting that $f'(x_2 + \alpha) < 0$ for all $x_2 > 0$ and $\Phi(x_2) \leq 1$, we have

$$T_1(\alpha) \leq -K \int_{x_2 > a_K} f'(x_2 + \alpha) dx$$

$$= Kf(a_K + \alpha) \leq Kf(a_K) = \frac{K}{\sqrt{2\pi}} e^{-a_K^2/2}.$$

For the term $T_2(\alpha)$, we have

$$|T_2(\alpha)| \leq K \sup_{x_2 \leq a_K} \Phi(x_2)^{K-1} \int_{x_2 \leq a_K} |f'(x_2 + \alpha)| dx_2$$

$$\leq K\Phi(a_K)^{K-1} \int_{x_2 \in \mathbb{R}} |f'(x_2)| dx_2 = \frac{2K}{\sqrt{2\pi}} \Phi(a_K)^{K-1}.$$

Using the classical estimate

$$\Phi(t) \leq 1 - \frac{1}{\sqrt{2\pi}} \frac{t}{t^2 + 1} e^{-t^2/2},$$

we obtain

$$\eta'(\alpha) \leq \frac{K}{\sqrt{2\pi}} e^{-a_K^2/2} + \frac{2K}{\sqrt{2\pi}} \Phi(a_K)^{K-1}$$

$$\leq \frac{K}{\sqrt{2\pi}} e^{-a_K^2/2} + \frac{2K}{\sqrt{2\pi}} \left( 1 - \frac{1}{\sqrt{2\pi}} \frac{a_K}{a_K^2 + 1} e^{-a_K^2/2} \right)^{K-1}.$$

By our choice of $a_K$, we have $e^{-a_K^2/2} = \frac{a_K \ln K}{cK}$, hence

$$\eta'(\alpha) \leq \frac{a_K \ln K}{c\sqrt{2\pi}} + \frac{2K}{\sqrt{2\pi}} \left( 1 - \frac{1}{\sqrt{2\pi}} \frac{a_K^2}{a_K^2 + 1} \frac{\ln K}{cK} \right)^{K-1}.$$

For $K$ large enough such that $cK/\ln K \geq \sqrt{e}$, we have $a_K^2 e^{a_K^2} \geq e$ and thus $a_K > 1$. Then

$$\frac{cK}{\ln K} = a_K e^{a_K^2/2} \geq e^{a_K^2/2}.$$

It follows that for $K$ large enough, we have

$$1 \leq a_K \leq \sqrt{2 \ln \left( \frac{cK}{\ln K} \right)}.$$

Consequently, for $K$ large enough and $\alpha > 0$,

$$\eta'(\alpha) \leq \frac{\ln K}{c\sqrt{\pi}} \sqrt{\ln \left( \frac{cK}{\ln K} \right)} + \frac{2K}{\sqrt{2\pi}} \left( 1 - \frac{1}{2\sqrt{2\pi}} \frac{\ln K}{cK} \right)^{K-1}.$$

Using the elementary inequality: for any $x > 1$,

$$(1 - \frac{1}{x})^x = \exp(x \ln(1 - 1/x)) = \exp(-x \sum_{k=1}^{\infty} x^{-k}/k) \leq e^{-1},$$

we obtain that, for any integer $K \geq 2$,

$$\frac{2K}{\sqrt{2\pi}} \left( 1 - \frac{1}{2\sqrt{2\pi}} \frac{\ln K}{cK} \right)^{K-1} = \frac{2K}{\sqrt{2\pi}} \left[ \left( 1 - \frac{1}{2\sqrt{2\pi}} \frac{\ln K}{cK} \right)^{\frac{2\sqrt{2\pi}cK}{\ln K}} \right]^{\frac{(K-1)\ln K}{2\sqrt{2\pi}cK}}$$

$$\leq \frac{2K}{\sqrt{2\pi}} e^{-\frac{(K-1)\ln K}{2\sqrt{2\pi}cK}} = \frac{2}{\sqrt{2\pi}} K^{1 - \frac{1}{2\sqrt{2\pi}c} + \frac{1}{2K\sqrt{2\pi}c}}$$

$$\leq \frac{2}{\sqrt{2\pi}} K^{1 - \frac{1}{2\sqrt{2\pi}c} + \frac{1}{4\sqrt{2\pi}c}} = \frac{2}{\sqrt{2\pi}} K^{1 - \frac{1}{4\sqrt{2\pi}c}}.$$

Now let us take

$$c = \frac{1}{4\sqrt{2\pi}}.$$

We obtain that, for $K$ large enough,

$$\eta'(\alpha) \leq \frac{\ln K}{c\sqrt{\pi}} \sqrt{\ln\left(\frac{cK}{\ln K}\right)} + \frac{2}{\sqrt{2\pi}}.$$

By the mean value theorem, we obtain

$$\eta(\alpha) \leq \left(\frac{\ln K}{c\sqrt{\pi}} \sqrt{\ln\left(\frac{cK}{\ln K}\right)} + \frac{2}{\sqrt{2\pi}}\right)\alpha.$$

This clearly implies the desired result. $\qquad\square$

Denote $\mu_{(1)} \geq \mu_{(2)} \geq \cdots \mu_{(K)}$ the non-increasing re-ordering of $\mu_1, \ldots, \mu_K$. We have

$$
\begin{aligned}
e_*(\alpha) &= \Pr\left(\mu_{(1)} - \mu_{(2)} < \alpha\right) \\
&= \Pr\left(\sigma_0^{-1}(\mu_{(1)} - \mu_{(2)}) < \sigma_0^{-1}\alpha\right).
\end{aligned}
$$

Then the theorem is a direct result of Lemma A.1.

## A.2 Proof of Proposition 4.1

Let $v$ be the midpoint between $\mu_k$ and $\mu_{i^*}$. Then

$$
\begin{aligned}
\Pr(\hat{\mu}_{k,n} \geq \hat{\mu}_{i^*,n} | \Delta_k, H_n) &= \Pr(\hat{\mu}_{i^*,n} > v)\Pr(\hat{\mu}_{k,n} \geq \hat{\mu}_{i^*,n} | \hat{\mu}_{i^*,n} > v) \\
&\quad + \Pr(\hat{\mu}_{i^*,n} \leq v)\Pr(\hat{\mu}_{k,n} \geq \hat{\mu}_{i^*,n} | \hat{\mu}_{i^*,n} \leq v) \\
&\leq \Pr(\hat{\mu}_{k,n} \geq v) + \Pr(\hat{\mu}_{i^*,n} \leq v) \\
&= \Pr(\hat{\mu}_{k,n} - \mu_k \geq \Delta_k/2) + \Pr(\hat{\mu}_{i^*,n} - \mu_{i^*} \leq -\Delta_k/2) \\
&\leq \exp\left[-\Delta_k^2/(8\tau_{k,n}^2)\right] + \exp\left[-\Delta_k^2/(8\tau_{i^*,n}^2)\right],
\end{aligned}
$$

where the last step is a direct result of the Gaussian tail bound shown in Appendix A.6.

## A.3 Proof of Theorem 5.1

Without loss of generality, we assume arm 1 is the optimal arm, i.e., $\mu_{i^*} = \mu_1$. Note that the initial round is $2K+1$, since every arm is pulled twice in the first $2K$ rounds. Denote

$$c_{k,t-1} = \sqrt{2\tau_{k,t-1}^2 \log n}.$$

We define the events that all confidence intervals from round $2K+1$ to round $n$ hold as,

$$\mathcal{E} = \left\{\forall k \in [K], t \in \{2K+1, \ldots, n\} : |\mu_k - \hat{\mu}_{k,t}| \leq \eta c_{k,t}\right\},$$

where $\eta = 1/(1+4\rho)$.

The error probability is decomposed as

$$
\begin{aligned}
\Pr\left(\hat{\mu}_{J_n,n} - \hat{\mu}_{1,n} > 0\right) &= \Pr\left((\hat{\mu}_{J_n,n} - \mu_{J_n}) - (\hat{\mu}_{1,n} - \mu_1) > \Delta_{J_n} | \mathcal{E}\right)\Pr\left(\mathcal{E}\right) \\
&\quad + \Pr\left((\hat{\mu}_{J_n,n} - \mu_{J_n}) - (\hat{\mu}_{1,n} - \mu_1) > \Delta_{J_n} | \bar{\mathcal{E}}\right)\Pr\left(\bar{\mathcal{E}}\right) \\
&\leq \Pr\left(\Delta_{J_n} < \eta(c_{1,n} + c_{J_n,n}) | \mathcal{E}\right) + \Pr\left(\bar{\mathcal{E}}\right), \quad (11)
\end{aligned}
$$

From (11), $e_n$ is decomposed into two terms. The first term is to compare the prior's gap with the upper confidence bounds. The second term is the probability that the confidence intervals do not hold.

At first, we focus on investigating the first term of the decomposition in (11). Denote $\beta = 1 + K^{-1}\sigma_0^{-2}\sigma^2$ and $\beta_1 = 1 + K^{-1}\sigma_0^2/(\sigma_0^2 + \sigma^2)$. Noting that $\tilde{c}_{k,n}$ just relies on its corresponding $T_{k,n}$, using $\tilde{c}_{k,n}$ instead of

$c_{k,n}$ breaks the dependence of arm $k$ on other arms. Because $\tilde{c}_{k,n}$ can be bounded by $c_{k,n}$: $c_{k,n} \leq \sqrt{\beta}\tilde{c}_{k,n}$ as shown in Zhu & Kveton (2022a). Thus, we have that

$$\Pr\left(\Delta_{J_n} < \eta(c_{1,n} + c_{J_n,n})\right) \leq \Pr\left(\Delta_{J_n} < \eta\sqrt{\beta}(\tilde{c}_{1,n} + \tilde{c}_{J_n,n})\right). \tag{12}$$

Now we bound $\Pr\left(\Delta_k \leq \eta\sqrt{\beta}(\tilde{c}_{1,n} + \tilde{c}_{k,n})|\mathcal{E}\right)$ for any $k \neq 1$. Denote $\Delta_{\min} = \min_{k \neq 1} \Delta_k$. We define the following event of comparing $\tilde{c}_{1,n}$ with $\Delta_{\min}$:

$$\mathcal{E}_1 := \{\tilde{c}_{1,n} \leq \Delta_{\min}/(\sqrt{\beta}(1 + \eta))\}.$$

We have that

$$
\begin{aligned}
&\Pr\left(\Delta_k < \eta\sqrt{\beta}(\tilde{c}_{1,n} + \tilde{c}_{k,n})|\mathcal{E}\right) \\
&\leq \Pr\left(\Delta_k < \eta\Delta_{\min}/(1 + \eta) + \eta\sqrt{\beta}\tilde{c}_{k,n}|\mathcal{E}_1, \mathcal{E}\right) + \Pr\left(\bar{\mathcal{E}}_1|\mathcal{E}\right) \\
&\leq \Pr\left(\Delta_k < \eta(1 + \eta)\sqrt{\beta}\tilde{c}_{k,n}|\mathcal{E}_1, \mathcal{E}\right) + \Pr\left(\bar{\mathcal{E}}_1|\mathcal{E}\right).
\end{aligned}
\tag{13}
$$

We shall show $\Delta_k \geq \eta(1 + \eta)\sqrt{\beta}\tilde{c}_{k,n}$ given $\mathcal{E}$ and $\mathcal{E}_1$ when $\eta$ satisfies $2\eta(1 + \eta)\sqrt{\beta/\beta_1} + \eta - 1 \leq 0$. In the following we take $\eta = 1/(1 + 4\rho)$. Lemma A.2 shows that on the event $\mathcal{E}$ the following result holds:

$$\Delta_k \geq (1 - \eta)\sqrt{\beta_1}\tilde{c}_{k,t} - (1 + \eta)\sqrt{\beta}\tilde{c}_{1,t},$$

implying that, on the events $\mathcal{E}$ and $\mathcal{E}_1$,

$$\Delta_k \geq (1 - \eta)\sqrt{\beta_1}\tilde{c}_{k,t}/2 \geq \eta(1 + \eta)\sqrt{\beta}\tilde{c}_{k,n},$$

where the second inequality is from $\eta$ satisfying $2\eta(1 + \eta)\sqrt{\beta/\beta_1} + \eta - 1 \leq 0$. Thus, (13) follows that

$$\Pr\left(\Delta_k < \eta\sqrt{\beta}(\tilde{c}_{1,n} + \tilde{c}_{k,n})|\mathcal{E}\right) \leq \Pr\left(\bar{\mathcal{E}}_1|\mathcal{E}\right) = \Pr\left(\Delta_{\min} < (1 + \eta)\sqrt{\frac{\beta\sigma_0^2\sigma^2\log n}{T_{1,n}\sigma_0^2 + \sigma^2}}|\mathcal{E}\right). \tag{14}$$

Now we investigate $T_{1,n}$. Lemma A.2 shows that on the event $\mathcal{E}$ for $k \neq 1$,

$$T_{k,t} \leq 2 + 2\Delta_k^{-2}(1 + \eta)^2\beta\sigma^2\log n.$$

It follows that

$$T_{1,n} = n - \sum_{k \neq 1} T_{k,n} \geq n - 2(K - 1)(1 + \eta)^2\Delta_{\min}^{-2}\beta\sigma^2\log n - 2(K - 1). \tag{15}$$

For decoupling $T_{1,n}$ and $\Delta_{\min}$, we define the following event of controlling the minimum gap:

$$\mathcal{E}_2 := \left\{\Delta_{\min} \geq 2(1 + \eta)\sqrt{(K - 1)\beta\sigma^2 n^{-1}\log n}\right\}.$$

On the event $\mathcal{E}_2$, (15) follows that

$$\sigma_0^2 T_{1,n} + \sigma^2 \geq \sigma_0^2 n/2 - 2\sigma_0^2(K - 1) + \sigma^2.$$

Thus, we have that

$$
\Pr\left(\Delta_{\min} < (1+\eta)\sqrt{\frac{2\beta\sigma_0^2\sigma^2\log n}{T_{1,n}\sigma_0^2+\sigma^2}}\Big|\mathcal{E}\right)
$$

$$
\leq \Pr\left(\Delta_{\min} < (1+\eta)\sqrt{\frac{2\beta\sigma_0^2\sigma^2\log n}{\sigma_0^2 n/2 - 2\sigma_0^2(K-1)+\sigma^2}}\Big|\mathcal{E}\right) + \Pr\left(\bar{\mathcal{E}}_2|\mathcal{E}\right)
$$

$$
= \Pr\left(\Delta_{\min} < (1+\eta)\sqrt{\frac{2\beta\sigma_0^2\sigma^2\log n}{\sigma_0^2 n/2 - 2\sigma_0^2(K-1)+\sigma^2}}\right)
$$

$$
+ \Pr\left(\Delta_{\min} < (1+\eta)\sqrt{2a(K-1)\beta\sigma^2 n^{-1}\log n}\right)
$$

$$
\leq c_K(1+\eta)\left(\sqrt{\frac{2\beta\sigma_0^2\sigma^2\log n}{\sigma_0^2 n/2 - 2\sigma_0^2(K-1)+\sigma^2}} + \sqrt{\frac{4\beta\sigma^2\log n}{n/(K-1)}}\right), \tag{16}
$$

where the last step is from Lemma A.1.

At last, we investigate the second term of the decomposition in (11). Define the following event: for each arm $k$

$$
\mathcal{E}_{3k} := \left\{|\bar{r}_{k,t} - \mu_k| \leq \eta\sqrt{\beta_1}\tilde{c}_{k,t}/2\right\}.
$$

From (4), we have that

$$
\Pr\left(\bar{\mathcal{E}}\right) \leq \sum_{k=1}^{K}\sum_{t=1}^{n}\Pr\left(|\hat{\mu}_{k,t}-\mu_k| > \eta c_{k,t}\right)
$$

$$
= \sum_{k=1}^{K}\sum_{t=1}^{n}\Pr\left(|\sigma^2/(T_{k,t}\sigma_0^2+\sigma^2)(\bar{r}_{0,t}-\mu_k) + w_{k,t}(\bar{r}_{k,t}-\mu_k)| > \eta c_{k,t}\right)
$$

$$
\leq \sum_{k=1}^{K}\sum_{t=1}^{n}\left[\Pr\left(\sigma^2/(T_{k,t}\sigma_0^2+\sigma^2)|\bar{r}_{0,t}-\mu_k| > \eta\sqrt{\beta_1}\tilde{c}_{k,t}/2\right) + \Pr\left(\bar{\mathcal{E}}_{3k}\right)\right]. \tag{17}
$$

where the last inequality is from $w_{k,t} < 1$ and $c_{k,n} \geq \sqrt{\beta_1}\tilde{c}_{k,n}$ as shown in Zhu & Kveton (2022a).

We shall investigate $\Pr\left(\sigma^2/(T_{k,t}\sigma_0^2+\sigma^2)|\bar{r}_{0,t}-\mu_k| > \eta\sqrt{\beta_1}\tilde{c}_{k,t}/2\right)$ and $\Pr\left(\bar{\mathcal{E}}_{3k}\right)$ respectively. From the properties of sub-Gaussian, $\bar{r}_{0,t} - \mu_0$ is sub-Gaussian with the parameters

$$
\nu_t =: \left[\sum_{k=1}^{K}(1-w_{k,t})T_{k,t}\right]^{-2}\sum_{k=1}^{K}(1-w_{k,t})^2 T_{k,t}^2(\sigma_0^2+\nu^2/T_{k,t}) \geq K^{-1}(\sigma_0^2+\nu^2/2),
$$

where the inequality is from the Cauchy–Schwarz inequality and $T_{k,t} \geq 2$. It follows that

$$
\Pr\left(\sigma^2/(T_{k,t}\sigma_0^2+\sigma^2)|\bar{r}_{0,t}-\mu_k| > \eta\sqrt{\beta_1}\tilde{c}_{k,t}/2\right) \leq 2\exp\left(-\frac{2\beta_1\sigma_0^2(T_{k,n}\sigma_0^2+\sigma^2)K\log n}{4(1+4\rho)^2\sigma^2(\sigma_0^2+\nu^2/2)}\right)
$$

$$
\leq 2\exp\left(-\frac{\beta_1\sigma_0^2 K\log n}{(1+4\rho)^2\sigma^2}\right), \tag{18}
$$

where the first step is from the sub-Gaussian tail inequality and the second step is from $T_{k,t} \geq 2$ and $\sigma^2 \geq \nu^2$

On the other hand, we have

$$
\Pr\left(\bar{\mathcal{E}}_{3k}\right) \leq 2\exp\left(-\frac{2T_{k,t}\eta^2\beta_1\sigma^2\sigma_0^2\log n}{4\nu^2(T_{k,t}\sigma_0^2+\sigma^2)}\right) \leq 2\exp\left(-\frac{\beta_1\sigma_0^2\log n}{\delta(1+4\rho)^2(2\sigma_0^2+\sigma^2)}\right),
$$

where the first step is from the sub-Gaussian tail inequality and noticing $\sigma^2 = \nu^2/\delta$, and the last step is from $T_{k,t} \geq 2$. Denote $m = \beta_1(1+4\rho)^{-2}\sigma^{-2}\sigma_0^2$, we have

$$\Pr\left(\bar{\mathcal{E}}\right) \leq 2Kn^{-mK+1} + 2Kn^{-\frac{\sigma^2 m}{\delta(2\sigma_0^2+\sigma^2)}+1}, \tag{19}$$

Therefore, combing (14), (16), and (17), the theorem is proved.

## A.4   Proof of Theorem 5.2

Let $\mu_{(1)} = \max\{\mu_k, k \in [K]\}$ and $\mu_{(K)} = \min\{\mu_k, k \in [K]\}$. We have that

$$\mathrm{sr}_n = \sum_{k \neq i^*} \Pr(J_n = k)\mathbb{E}\left[\mu_* - \mu_{J_n} \mid J_n = k\right]$$
$$\leq e_n\mathbb{E}\left[\mu_{(1)} - \mu_{(K)}\right] \leq 2\sigma_0\sqrt{2\log K}e_n,$$

where the first inequality is from $\mu_{J_n} \geq \mu_{(K)}$, and the last step is due to the fact of $\mathbb{E}\left[\mu_{(1)} - \mu_{(K)}\right] \leq 2\sigma_0\sqrt{2\log K}$. Combing Theorem 5.1, the proof is concluded.

## A.5   Lemmas

**Lemma A.2.** *On the event $\mathcal{E}$, We have the following two results: for $k \neq 1$,*

$$T_{k,t} \leq 2 + 2\Delta_k^{-2}(1+\eta)^2\beta\sigma^2\log n \tag{20}$$
$$(1-\eta)\sqrt{\beta_1}\tilde{c}_{k,t} \leq (1+\eta)\sqrt{\beta}\tilde{c}_{1,t} + \Delta_k. \tag{21}$$

*Proof.* (20) is obviously true at time $t = 2K + 1$. We assume that it holds at time $t \geq 2K + 1$. If $I_{t+1} \neq k$, then $T_{k,t+1} = T_{k,t}$, thus it still holds. If $I_{t+1} = k$, it means that $\hat{\mu}_{k,t} + c_{k,t} \geq \hat{\mu}_{1,t} + c_{1,t}$. Note that on $\mathcal{E}$, we have that

$$\hat{\mu}_{1,t} + c_{1,t} \geq \mu_1, \text{ and } \hat{\mu}_{k,t} + c_{k,t} \leq \mu_k + (1+\eta)c_{k,t}.$$

They follows

$$(1+\eta)c_{k,t} \geq \Delta_k.$$

Thus, $T_{k,t} \leq 2\Delta_k^{-2}(1+\eta)^2\beta\sigma^2\log n$ holds due to $\tau_{k,t}^2 \leq \beta\sigma_0^2\sigma^2/(T_{k,t}\sigma_0^2 + \sigma^2)$ shown in Lemma A.3. By using $T_{k,t+1} = T_{k,t} + 1$, we prove (20).

(21) is obviously true at the initial time $t = 2K + 1$. We assume that it holds at time $t \geq 2K + 1$. If $I_{t+1} \neq 1$, then $T_{1,t+1} = T_{1,t}$, thus it still holds. If $I_{t+1} = 1$, it means that

$$\hat{\mu}_{k,t} + c_{k,t} \leq \hat{\mu}_{1,t} + c_{1,t}.$$

Note that on $\mathcal{E}$, we have that

$$\hat{\mu}_{1,t} + c_{1,t} \leq \mu_1 + (1+\eta)c_{1,t}, \text{ and } \hat{\mu}_{k,t} + c_{k,t} \geq \mu_k + (1-\eta)c_{k,t}.$$

They follow

$$(1-\eta)c_{k,t} \leq (1+\eta)c_{1,t} + \Delta_k.$$

Since $c_{k,t} \geq \sqrt{\beta_1}\tilde{c}_{k,t}$ and $c_{1,t} \leq \sqrt{\beta}\tilde{c}_{1,t}$ shown in Lemma A.3, we have that

$$(1-\eta)\sqrt{\beta_1}\tilde{c}_{k,t} \leq (1+\eta)\sqrt{\beta}\tilde{c}_{1,t} + \Delta_k.$$

By using $T_{k,t+1} = T_{k,t} + 1$, we prove (21). $\qquad\square$

**Lemma A.3.** *(Lemmas 1 & 5 in Zhu & Kveton (2022a))*

$$\frac{\sigma_0^2\sigma^2}{T_{k,t}\sigma_0^2 + \sigma^2}(1 + K^{-1}\sigma_0^2/(\sigma_0^2 + \sigma^2)) \leq \tau_{k,t}^2 \leq \frac{\sigma_0^2\sigma^2}{T_{k,t}\sigma_0^2 + \sigma^2}(1 + K^{-1}\sigma^2\sigma_0^{-2}).$$

### A.6 Some Properties

**Fact 1** (Gaussian tail bound) Let $X$ be a Gaussian random variable, i.e., $X \sim \mathcal{N}(0, \sigma^2)$, then for all $\alpha > 0$,

$$\Pr(X \geq \alpha) \leq \exp\left(-\frac{\alpha^2}{2\sigma^2}\right).$$

**Fact 2** (Joint distribution of ordered statistics) Denote $F(x) = \mathrm{P}(X \leq x)$ and $f(x)$ as its density. Let $X_{(1)} \geq X_{(2)} \geq \cdots \geq X_{(K)}$. For $x_1 \geq x_2$, the density of $(X_{(1)}, X_{(2)})$ is

$$p(x_2, x_1) = n(n-1)F(x_2)^{n-2}f(x_2)f(x_1).$$

**Fact 3** (Some results on Gaussian)

$$\frac{1}{\sqrt{2\pi}}\frac{t}{t^2+1}\exp(-t^2/2) \leq 1 - \Phi(t) = \Pr(X > t) \leq \frac{1}{t\sqrt{2\pi}}\exp(-t^2/2).$$

$$\Pr(X_{(2)} < t) = [\Phi(t)]^n + n[1 - \Phi(t)][\Phi(t)]^{n-1}.$$

### A.7 More Results of Experiments

**Performance of the two-stage algorithm** We show the empirical studies of the two-stage algorithm. For overall checking the performance, we check it under various $q = 0.1, 0.2, \ldots, 0.9$, we report their performance under Setup F4 in Figure 3. The figure shows the Two-Stage algorithm is bad under various $q$.

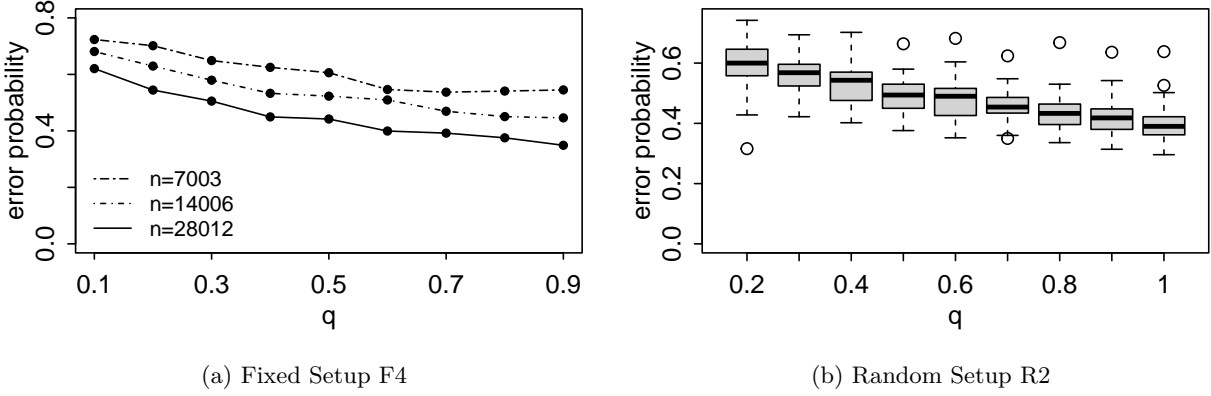

(a) Fixed Setup F4          (b) Random Setup R2

Figure 3: The error probability of the Two-Stage algorithm with various $q$ values, averaged over 1000 independent executions (results in standard deviations of less than 0.016).

**Impact of arm number $K$.** We ran the experiments with $n = 20, 40, 80$ arms in order to examine how the performance of each algorithm scales as the number of arms grow. We report the result on the arithmetic setting (Setup F4) in Figure 4, where $K = 40$ and $80$ are shown. Comparing various $K$, the benefit of using RUE increases as $K$ increases.

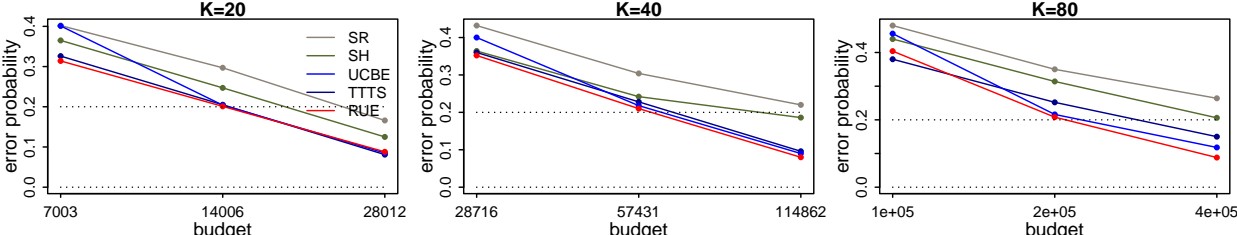

Figure 4: The error probability of the different algorithms in Setup F4 with more arms, 40 and 80 arms (left and right subfigures respectively). The results are averaged over 1000 independent executions (all standard errors are less than 0.016 and not reported). For $K = 80$, we set $N = 400000$ as the maximal budget for the limit of resources when $2H$ is too big.

