# OpenReview forum: "UCB Exploration for Fixed-Budget Bayesian Best Arm Identification"
_TMLR — Accepted by TMLR_

### Review · Reviewer_VTwj · 2024-05-03

**Summary Of Contributions:**

The paper studies best-arm identification (BAI) in the fixed budget setting with Bayesian assumptions on the mean arm rewards; specifically, they assume the means are themselves sampled from a Gaussian distribution. The paper claims that, for existing BAI fixed budget algorithms with exponentially decaying failure probability bounds, these bounds are actually instance-dependent and may be only polynomially decaying in some instances. The paper develops an algorithm called RUE (Random Effects UCB Exploration) which may be seen as a conversion of a previous ReUCB (Zhu & Kveton, 2022) algorithm for multi-armed bandits (under the Bayesian assumptions) to the BAI setting. The paper provides upper bounds on the failure probability and simple Bayes regret for RUE and shows that they are both of order $\mathcal{O}(\sqrt{K/n})$ up to log factors. RUE is compared to previous (non-Bayesian) BAI algorithms and to one Bayesian BAI algorithm.

**Audience:**

Yes

**Broader Impact Concerns:**

None.

**Claims And Evidence:**

Yes

**Requested Changes:**

### Would strengthen work:
1. Please answer the questions in the previous section, and add the discussion to the paper for clarity.
2. Compare the algorithm to the Bayesian BAI algorithms in Russo (2016), or justify why they are not comparable.

**Strengths And Weaknesses:**

### Strengths:
1. Important theoretical results regarding the proposed algorithm are provided.
2. The proposed algorithm is empirically evaluated.
3. The writing and exposition is generally clear and easy to understand.

### Weaknesses/Questions:
1. The paper claims that UCBE in the ordinary fixed budget BAI setting is instance dependent because in some instances, when the problem complexity $H$ is greater than some threshold dependent on $K$ and $n$, the exponentially decaying bound becomes polynomial. However, the provided bounds for RUE are also polynomial. Is the factor of $\sqrt{K}$ improvement the best we can do with the additional side information of the prior? Are there some instance specific conditions where we can recover the exponential bound? In principle, we should be able to do better with more information and assumptions.
2. How does the work in this paper compared to the Bayesian BAI methods in "Simple Bayesian Algorithms for Best-Arm Identification" by Russo (2016) ? Are the theoretical results comparable? Why weren't the algorithms from this paper evaluated in the empirical studies? It seems to be closer to the studied setting than UCBE and the other non-Bayesian BAI algorithms.

---

> ### Author Response · Authors · 2024-08-14
>
> Thank you for constructive comments to improve our manuscript. We have carefully addressed each comment.
>
> -  on the exponentially decaying bound
>
> To answer this question, we have considered a two-arm bandit problem and examined a scenario with full information, where both arms are observed each time. This analysis demonstrates that achieving exponentially decaying regrets in fixed-budget BAI is infeasible when the gap is small, and we have clarified the aim of our paper accordingly.  The aim of this paper is not to develop an algorithm with exponentially decaying bounds, but rather to create an algorithm that significantly addresses the small-gap issue and achieves polynomially decaying regret bounds.  We have added more discussion details on the requirement for $\sigma_0^2$ in Section 5.1.
>
> - on comparison to TTTS
>
> We have added a comparison to the TTTS proposed by Russo (2016). The results show that our method RUE outperforms TTTS in most cases but performs slightly worse in some cases.

---

### Review · Reviewer_RtAM · 2024-05-03

**Summary Of Contributions:**

This paper presents a Bayesian algorithm for best-arm identification, based on a variation of the Upper Confidence Bound method, and then bounds the resulting error probability and regret (in a Bayesian sense). The method is compared with existing methods numerically, and is shown to perform well.

**Audience:**

Yes

**Claims And Evidence:**

No

**Requested Changes:**

* Explain more carefully the effect of small $\sigma_0$, and determine whether this method / analysis actually gives any advantage over the prior gap-dependent bounds.
* Specifically fix the incorrect language about the $\tilde O(\sqrt{K/n})$ bounds.
* Fix the math errors.
* Update the literature review.

**Strengths And Weaknesses:**

# Strengths

* The writing is well-structured.
* The math appears to be mostly correct, and the mistakes appear to be minor or correctable.
* The general method is reasonably interesting

# Weaknesses

* The main motivation, which is that gap-based error bounds become trivial when gaps are small, is not really solved by this method or analytic framework. Similar trivial bounds result from this work as well, when $\sigma_0$ is small. (More on this below.)
* The math has some fairly basic math errors, some of which appear to be simple mis-statements, others appear to be conceptual errors.
* In the proof outline from the main text, the notation $\tilde c_{k,n}$ is used, but the term is not actually defined till the appendix.
* The literature review is likely out of date. There are no references after 2022, and this is a fast-moving field. I would imagine that many relevant works have been published in the past two years. In order to maintain anonymity, I purposely did not do a literature review in case this paper exists as a preprint.

## Trivial Bounds for Small $\sigma_0$

The right side of the bound in Theorem 5.1 goes to $\infty$. In particular, as  $\sigma_0\to 0$, we get the following limits:
* $\gamma \sqrt{H_b}\to \infty$
* $Kn^{-mK+1}\to Kn$

So, it suffers from similar problems as the instance-dependent bounds (which depend on gap calculations)

Similarly, the regret bound of Theorem 5.2 is trivial for small $\sigma_0$. (It goes to $\infty$ for the same reasons.)

As a result, the discussion throughout the paper about $\tilde O(\sqrt{K/n})$ bounds appears to be incorrect, or at best, rather misleading.

## Incorrectness of Proposition 4.1

The most glaring mathematical error is Proposition 4.1. The right side of the probability expression is a random variable, depending on the number of times the arms have been pulled. So, this is not a sensible expression. The only way to make sense is to be examining some sort of conditional probability, such as conditioning on the arm pull counts.

Luckily, it appears that Proposition 4.1 is not actually used in the main analysis.

## Smaller Math Errors

- In the use of sub-Gaussian concentration bounds, one-sided bounds appear to be used for two-sided expressions, in many cases. Most likely, these bounds need to increase by a factor of 2.
- The event $\mathcal{E}_3$ should really be a collection events for each arm, $k$.

---

> ### Author Response · Authors · 2024-08-14
>
> Thank you for very helpful comments for improving this manuscript.
>
> - on small gaps vs small $\sigma_0^2$
>
> In the revised version, we have focused on the minimum gap $\Delta_{min}$ in Section 2,
> as  the optimality of UCBE particularly depends on $\Delta_{min}$.
> This leads to the discussion on the relationship between $\sigma_0^2$ and $\Delta_{min}$ in classical BAI at the end of Section 3: When $\Delta_{min}$ is very small, $\sigma_0^2$ can be $O(1)$.
>
> - the impact of small $\sigma_0^2$ on the bounds
>
> We have outlined the requirement for $\sigma_0^2$ in Section 5.1, where we allow $\sigma_0^2=O(1/K)$, meaning it can be small as $1/K$. When $\sigma_0^2=O(1/K)$, $H_b$ remains $O(K)$ given $\sigma^2=O(1)$.  Therefore, the bounds remain $\tilde{O}(\sqrt{K/n})$ when $\sigma_0^2=O(1/K)$.
>
> - on the incorrectness of Proposition 4.1
>
> It's a conditional probability given $\Delta_k$ and $H_n$. In the original version, we mentioned that it's given $\Delta_k$ and $H_n$
>  but did not provide the explicit form of the conditional probability.
> We have now corrected it.
> Additionally, we have modified other math errors.  Thank you so much for pointing these errors.
>
> - on new relevant works
>
> We have added some new papers and related to our paper. Please check the third paragraph in Section 7.

---

### Review · Reviewer_aRGJ · 2024-08-10

**Summary Of Contributions:**

This paper presents a novel method for fixed-budget Bayesian best-arm identification (BAI). The paper proposes a new perspective on Bayesian BAI, in which the mean reward of all arms are sampled i.i.d. from a Gaussian distribution with a fixed variance. The observed rewards are corrupted by a sub-Gaussian noise. Analysis of both the failure probability and expected simple regret are given, and intuitive comparisons are made w.r.t. prior BAI works. The main theoretical advantage of the theoretical results in this work is that the failure probability is exponentially decaying even for small reward gaps (in other words, small variance of mean rewards). Experiments are performed, for both random and non-random mean rewards, to show that the propose method works competitively.

**Audience:**

Yes

**Broader Impact Concerns:**

No ethical concerns since the paper is more theoretical.

**Claims And Evidence:**

Yes

**Requested Changes:**

- In addition to what I've discussed above, another interesting thing about the paper is that the algorithm is in fact almost the same as classical UCB (except for a small difference in the exploration parameter). So, does this mean that UCB can both achieve BAI and cumulative regret minimization? I think some discussions would be helpful.

**Strengths And Weaknesses:**

Strengths:
- The novel framework of Bayesian BAI (in which the mean rewards of different arms are sampled from a normal distribution) is interesting and makes sense to me. It reminds me of an assumption commonly used in Bayesian optimization: people often assume that the reward function is sampled from a Gaussian process, and also assume that the noise is Gaussian or sub-Gaussian (it would be great if you discuss the connections with Bayesian optimization from this perspective).
- For the experiments, I particularly like the ones with fixed mean rewards $\mu_k$, since it potentially shows the performance of the method in practical scenarios where the Bayesian assumption is likely to be violated.

Weaknesses:
- As the authors also commented in the Conclusion section, the proposed method requires the mean reward variance $\sigma_0^2$ to be known. But I also understand that such assumptions are not uncommon in the bandit literature, and that the paper discussed methods to estimate it when it's unknown. So this may not be a major concern.
- Section 5.1, first line: a formula for $\delta$ is given here, which is required for the failure probability and simple regret to be exponentially decreasing. What's the intuition behind this formula? When is this formula likely to be satisfied? I think some discussions are needed.
- Also in Section 5.1, although I agree that the failure probability and simple regret can be exponentially decreasing for very small reward gaps, however, if I understand correctly, it's hard to characterize how small the reward gap can be. This is because the final theoretical results depend on $\sigma_0$. The consequence is that making comparisons with classical BAI (and justifying the theoretical advantage of the proposed method) is not so straightforward.
- In the experiments in Section 6.1, why do you draw the $\mu_k$'s from a uniform distribution? Why not a Gaussian distribution as in your assumptions?
- In your experiments, did you use the estimated $\sigma_0^2$ or the groundtruth one?
- The figure legends are missing in Figure 1.

---

> ### Author Response · Authors · 2024-08-14
>
> Thank you for numerous constructive comments to enhance our manuscript. We have carefully addressed each comment. Below we describe the changes among them.
>
> - relation to Bayesian optimization
>
> Yes, our Bayesian setup is related to Bayesian optimization. Both use Gaussian prior, but in our Bayesian BAI setting, the Gaussian prior models reward means of individual arms.  We have added a paragraph on this in Section 7 (Relation Work).
> Additionally, this comment reminds me a potential for using our proposed algorithm for hyper-parameter optimization,
> which we have discussed in Section 8.
>
> - on $\delta$ in Section 5.1
>
> We have provided additional explanation and  intuition in Section 5.1.
> Specifically, $\delta$ is the ratio of $\nu^2$ to the variance $\sigma^2$ of the Gaussian likelihood. This is from the algorithm part for setting the Gaussian likelihood. The setup is analogous to the one in Agrawal & Goyal (2013).
> We have included this information in Section 4.1.
>
> - on $\sigma_0^2$
>
> We have outlined the requirement on $\sigma_0^2$ in Section 5.1,
> where we allow $\sigma_0^2=O(1/K)$, meaning it can be small.
> And we discuss the relationship between $\sigma_0^2$ and small gap in classical BAI at the end of Section 3: When $\Delta_{min}$ is very small, $\sigma_0^2$ can be $O(1)$.
>
> - on experiments
>
> We have supplied these informations.
> (1) In R3, we used a uniform distribution  to evaluate the performance of our proposed algorithm when the Gaussian assumption does not hold.
> (2) In the implementation of RUE, we used the estimates of these variances.
> (3) We have added the legend to figure 1.

---

### Author Response · Authors · 2024-10-03
**Correct a typo**

There is a typo in the definition of $e_*(\alpha)$ in Section 3.
It should be $e_*(\alpha)=\text{Pr}\left(\mu_*-\sup_{k\neq i^*}\mu_k\leq \alpha\right)$.

---

### Decision · Action_Editor_7U99 · 2024-09-18

**Recommendation:** Accept with minor revision

**Comment:**

The paper presents a novel approach to fixed-budget Bayesian Best-Arm Identification (BAI), with significant theoretical contributions and experimental validation. Initially, the reviewers had concerns regarding mathematical rigor, theoretical explanations, and comparisons to prior work. However, after the authors' detailed and well-received rebuttal, all three reviewers now recommend acceptance, highlighting the value of the work and acknowledging that the issues raised have been satisfactorily addressed.

While the paper is in good shape, minor revisions are still needed. Specifically, some adjustments to the use of mathematical notation (such as replacing Big-O with Big-Omega where appropriate) and a correction to the definition of the supremum in the probability expression are required. Additionally, a few formatting issues, such as appendix numbering, should be addressed for clarity.

With these small revisions, the paper is ready for publication and is a valuable contribution to the bandit community.

**Audience:**

Yes, the findings will interest TMLR's audience, especially in Bayesian methods and multi-armed bandit problems.

**Claims And Evidence:**

Yes, the claims in the submission are supported by accurate and convincing evidence. The theoretical results and experimental validations are clear and well-supported. The authors have addressed earlier concerns, and with a few minor revisions, the evidence backing the claims is solid.